# Are men's gender equality beliefs associated with self-reported intimate partner violence perpetration? A state-level analysis of California men

**Kalysha Closson**[1,2]\*, **Nicole E. Johns**[1], **Anita Raj**[1,3,4]

**1** Center for Gender Equity and Health, University of California, San Diego, California, United States of America, **2** Faculty of Health Sciences, Simon Fraser University, Burnaby, British Columbia, Canada, **3** Newcomb Institute, Tulane University, New Orleans, Louisiana, United States of America, **4** Tulane School of Public Health and Tropical Medicine, New Orleans, Louisiana, United States of America

\* kvenditt@sfu.ca

## Abstract

### Objectives

To assess the association between gender equality beliefs and self-reported intimate partner violence (IPV) perpetration among California men.

### Methods

We analyzed men's data (N = 3609) from three waves (2021, 2022, and 2023) of cross-sectional data from a statewide sample of California adults. We assessed gender equality beliefs using a three-item measure adapted from the World Values Survey, with higher scores representing more gender unequal beliefs (e.g., 'On a whole, men make better political leaders than women'). We assessed IPV perpetration in the last year by asking a) whether men committed any form of violence in the last year (physical violence, use or threat of violence with a weapon, sexual violence) and b) among those who reported committing violence, who they committed violence against. Those reporting violence against "a spouse or romantic partner" were categorized as perpetrating past-year IPV. Analyses were weighted to yield population estimates. Crude and adjusted logistic regression models evaluated the association between gender equality beliefs and past-year IPV perpetration.

### Results

Almost 2% of men–equivalent to more than 280,000 men in California—reported IPV perpetration in the past year [1.9% (95%CI = 1.4–2.5)], and every one-point increase in their gender equality belief scale score [indicative of less gender equitable beliefs] was associated with 2.1 times greater odds of perpetrating past-year IPV (AOR: 2.14, 95%CI 1.61–2.86).

**Data Availability Statement:** The data used for this study is publicly available via the Open ICPSR repository. 2021: https://www.openicpsr.org/openicpsr/project/204402/version/V1/view 2022:

https://www.openicpsr.org/openicpsr/project/204401/version/V1/view 2023: https://www.openicpsr.org/openicpsr/project/199087/version/V1/view The code for this analysis is available as a Supporting Information file.

**Funding:** This work was supported by Blue Shield of California Foundation (https://blueshieldcafoundation.org) #COV-2207-18216 (AR) and Bill and Melinda Gates Foundation (https://www.gatesfoundation.org/) #INV002967 (AR). The funders had no role in the study design, data collection or analysis, decision to publish, or preparation of the manuscript.

**Competing interests:** The authors have declared that no competing interests exist.

## Conclusions

Findings support prior research indicating that patriarchal beliefs reinforce men's violence against women in relationships and signal a need for violence prevention efforts focused on boys and men to that can include normative belief shifts related to women's capacities.

## Introduction

More than 2 in every 5 adult women in the United States (U.S) has been physically or sexually abused or threatened by an intimate partner in their lifetime, and more than 72% of these women had their first experience of intimate partner violence (IPV) before the age of 25 years [1]. IPV increases women risk for a broad array of health concerns, including STIs/HIV, unwanted pregnancy, depression and anxiety [2–4], and it remains the leading cause of death for women in pregnancy in the U.S. [5]. Further, while both women and men can be victims of IPV, women are more likely than men to face greater severity of violence and risk of death, as indicated by U.S. homicide data from 2021 revealing that 34% of women homicide victims, compared with 6% of men homicide victims, were killed by a partner [6]. Understanding factors associated with IPV perpetration by men is important to support evidence-based IPV prevention programming tailored to men.

Studies on men's IPV perpetration in the U.S. demonstrate an array of risk factors associated with IPV. A strong body of evidence exists demonstrating individual-level emotional and behavioral risk factors associated with IPV, including depression, emotion regulation difficulties, substance use, aggressive or violence against non-partners (e.g., bullying, community violence), and other antisocial activities such as criminal behavior or truancy [7–12]. Men who have been victims of violence in childhood or adulthood, including from IPV, and those with high levels of community or institutional (e.g., prison) violence exposure are also more likely to perpetrate IPV, possibly due to the normalization of violence but also as a response to trauma [11, 13, 14]. Ecological factors, including the neighborhood environment and perceptions of one's neighborhood have also been highlighted as important determinants of IPV perpetration among men [15]. Traditional masculinity gender role expectations have also been linked to IPV perpetration and attitudes of acceptability of IPV, particularly among racial/ethnic minority men in the U.S. [14, 16–19]. Among racialized men, experiences of discrimination and resultant depression can contribute to masculinity discrepancy stress, which occurs when men feel they have fallen short of achieving expected masculinity roles or norms (e.g., employment and high-income generation, heterosexuality, sexual engagement). Evidence suggests that this too is a risk factor for men's perpetration of IPV [20–22].

While many studies offer much insight into social, behavioral, and even gendered risk factors for men's IPV perpetration, they offer little insight into what role if any men's individual attitudes toward women have toward IPV perpetration. Global efforts emphasize respect for women as key for prevention of men's violence against women including IPV [23], and U.S. IPV prevention efforts with youth similarly ground health relationships education in notion of respect and gender equality [24]. Thus, while we have data understanding this connection at a society-level connection, we lack data in understanding how individual men's views of women and gender equality correspond with individual perpetration of IPV. Recent research suggests national and population-based variability in gender equality attitudes, with men holding less gender equitable beliefs than women, and states with more progressive policies for women (e.g., access to abortion) seeing more gender equitable beliefs relative to states with less

progressive policies for women [25]. This issue requires further exploration, as gender-based violence, including IPV, is fundamentally rooted in pervasive societal gender inequalities [26].

Gender inequalities remain a global concern [27, 28], and the U.S. ranks lower than many developed countries on multiple aspects of gender equality, including women's leadership and political positioning [29]. Ecological analyses demonstrate that such inequalities are connected to higher rates of IPV in the U.S. [30], but individual level analyses on this issue remain lacking [31]. This paper aims to fill these gaps by examining the association between gender equality ideologies related to women in leadership and perpetration of IPV, among a statewide sample of men in California. Findings from this work can extend prior research, described above, to offer insight into whether we need to expand our gendered approaches to IPV prevention by increasing gender equality ideologies and men's value of and respect for women, broadly as well as in their relationships.

## Methods

### Data source

Data for this study comes from three waves of the California Violence Experiences Survey (CalVEX) conducted in 2021, 2022, and 2023. The CalVEX Survey is an annual online survey of adult (age 18+) residents of California that collects information on experiences of violence, discrimination, and several additional related factors.

The survey research firm NORC at the University of Chicago conducts CalVEX each year, utilizing both probability and non-probability samples to survey at least 2000 residents annually. NORC then performs statistical calibration to combine the probability and non-probability sample to produce a survey-weighted sample that is representative of the non-institutionalized adult population of the state. Each year, we verified representativeness of the weighted data via comparison with census and other state estimates of key demographic characteristics including gender, age, income, race and ethnicity, education, employment, disability, citizenship, and sexual identity. Additional details on the survey methodologies used for CalVEX data collection and survey weighting have been published elsewhere [32–35]. For the probability sample, the survey response rate and weighted American Association for Public Opinion Research (AAPOR) Response Rate 3 (RR3) cumulative response rates in 2021, 2022, and 2023 were 27.8%/4.1%, 27.3%/3.5%, and 32.9%/5.4%, respectively. These response rates are similar to those suggested by other online survey panels [36].

In 2021, participants were recruited from March 12th to March 24th, in 2022, between March 16th and March 31st, and in 2023 between March 27th and May 30th. For each year of the CalVEX survey (2021–2023) participants provided written consent to NORC for inclusion at the time of panel enrollment. Participants were able opt out of the survey panels as well as the CalVEX survey at any time. All study procedures were reviewed and approved by IRBs at the University of Chicago and University of California San Diego.

Data for this study are limited to men; women and non-binary, genderqueer, gender fluid, or prefer to self-describe gender identity individuals are excluded.

### Primary outcome

The primary outcome is self-reported perpetration of intimate partner violence (IPV) within the past year. All CalVEX respondents are asked whether they have *experienced* each form of physical or sexual violence within the past year, and separately asked whether they have *committed* that form of violence within the past year. Assessment of violence perpetration includes physical violence (inclusive of physical abuse and knife and gun violence), and sexual harassment and assault (inclusive of verbal harassment, homophobic or transphobic harassment,

physically aggressive harassment, quid pro quo harassment or coercion, and sexual assault). For those who have committed any form of violence within the past year, they are asked who they committed that violence against, including the answer option "A spouse or romantic partner". Self-reported perpetration of IPV was indicated when a respondent reported committing any form of physical or sexual violence in the past year and reporting that this violence was against a spouse or romantic partner.

## Primary exposure

The primary predictor of interest is gender equality beliefs. This measure was captured via three items reflecting gender equality beliefs in the World Values Survey (WVS) [37]. The three items were presented with the prompt "Now please tell me whether you agree or not with the following statements about women": "1. On the whole, men make better political leaders than women", "2. A university education is more important for men than for women", "3. On the whole, men make better business executives than women do". Answer options were a 5-item Likert scale Agree (5), Somewhat agree (4), Neither agree nor disagree (3), Somewhat disagree (2), or Disagree (1).

This three-item measure has not been previously validated. Given the limited length of the CalVEX survey (15 minutes), we selected three items from the WVS that focused on women in leadership. These items aimed to capture gender roles outside the household. The selection process was based on expert opinion, including consulting with CalVEX advisory board members and experts in measurement of gender norms and attitudes. For the analyses presented here, we considered inclusion of each item separately or a three-item scale; due to high correlation between items ($R = 0.7$–$0.8$) and high scale reliability ($\alpha = 0.90$), we ultimately present the three-item scale. The scale ranges from 1 (Disagree with all items, more gender equal beliefs) to 5 (Agree with all items, less gender equal beliefs).

## Additional variables of interest

We examine a number of additional factors which are potentially associated with IPV perpetration and/or gender equality beliefs.

**Mental health symptoms and substance use.** We included one measure of depression/anxiety symptoms in the prior 2 weeks using the 4-item Patient Health Questionnaire (PHQ4) scale [38]. Respondents were classified as having normal, mild, moderate, or severe depression and/or anxiety symptoms as defined by the PHQ4 measure, which we dichotomized to normal/mild or moderate/severe symptoms. We include two indicators of substance misuse, including an indicator of six or more days of binge drinking in the past month, and an indicator of six or more days of illicit drug use or use of a prescription drug without a prescription in the past month.

**Neighborhood safety.** Perceived neighborhood safety was asked as "How safe do you think your neighborhood is from violence and crime?" with answer options Extremely safe, Quite safe, Slightly safe, and Not at all safe. Slightly and Not at all safe were combined due to response distributions for a three-level measure for analyses.

**Gun ownership.** Current gun ownership was assessed directly, with a yes or no response option.

## Sociodemographic characteristics

We present rates of self-reported IPV perpetration by a number of sociodemographic characteristics and include these in adjusted regression analyses. These factors were selected *a priori* based on established association with gender equality beliefs [39] and violence perpetration in

prior literature and availability in the CalVEX survey data [8, 34, 35, 40–47]. These characteristics are: age (18–24, 25–34, 35–44, 45–54, 55–64, 65–74, 75+; included as continuous years in adjusted models), race/ethnicity (White, Hispanic/Latinx, Black/Asian/Other/Multiple races), ideology (Liberal, Moderate, Conservative), education (Less than high school diploma/GED, Completed high school/Some college/Associate's degree, Bachelor's degree/4-year college degree, Graduate degree), household income (Less than $60,000, $60,000–100,000, More than $100,000), marital status (Married or living with partner, Not married or living with partner), and sexual orientation (Gay/Bisexual/Other self-described, Heterosexual). We also present rates of self-reported IPV perpetration by year (2021, 2022, 2023) and include year in all adjusted analyses.

## Statistical analysis

We first present reported rates of self-reported IPV perpetration, overall and by specific form of violence. We also present state population-level estimates of perpetration based on census population estimates.

Next, we present bivariate comparisons of self-reported IPV perpetration with gender equality beliefs and with all other predictors of interest, utilizing Pearson's chi-squared tests to assess association.

Finally, we present a series of unadjusted and adjusted multivariate logistic regression analyses to assess the association between self-reported IPV perpetration and gender equality beliefs when accounting for sociodemographic and other plausibly associated factors. All adjusted models include the gender equality beliefs scale score and year. The models additionally controlled for:

1. Sociodemographic factors

2. Sociodemographic factors + mental health symptoms and substance use, neighborhood safety, and gun ownership

All sociodemographic factors noted above as well as mental health symptoms, substance use, neighborhood safety, and gun ownership were considered for inclusion in adjusted models; due to evidence of multicollinearity (VIF >6), education was removed; VIFs were <2 for all remaining factors and thus all other variables were retained in adjusted models.

As a post-hoc sensitivity analysis, we examine self-reported IPV perpetration stratified by sexual orientation, e.g., separately examine rates of perpetration and gender equality beliefs scale score among heterosexual vs gay, bisexual, or another non-heterosexual sexual orientation (GBO) men. We also replicate unadjusted and sociodemographic-adjusted models examining the association between self-reported perpetration and gender equality beliefs, stratified by sexual orientation.

All analyses accounted for survey weighting when calculating rates and estimates of association. Weighted percentages and unweighted Ns are presented. Statistical significance was set at p<0.05 for all comparisons, and 95% confidence intervals (CIs) are reported throughout. All analyses were conducted using Stata 15.1.

## Ethical review

All research procedures were approved by both NORC via the University of Chicago and the University of California, San Diego Institutional Review Boards (IRB) (UCSD Project #806718). Participants provided written informed consent when initially enrolling in a survey panel. Participants were able to skip or decline questions and were able to stop the survey at any point. Given the sensitive nature of survey items, information including web links to

victim services and a hotline number were provided at the bottom of the survey web screen. All data were de-identified by NORC prior to provision of the data to the UCSD team for analysis.

# Results

## Characteristics of the sample

The CalVEX survey sample includes 3,658 men across the three survey years. Men who declined response to one or more of the misogynistic attitude items (n = 49) were excluded from analyses; no men declined to answer the set of items from which we derived self-reported IPV perpetration. Our final analytic sample included N = 3,609 men (n = 1,153 from 2021, n = 1,004 from 2022, and n = 1,452 from 2023).

Men were 48 years old on average (mean 47.6 years, SD 17.5 years). 41% were White, 33% were Hispanic/Latino, and 26% were Black, Asian, or other/multiple races. About half of men identified as moderate (48%) with 28% identifying as liberal and 24% as conservative. Most (89%) had completed high school, and 19% had a graduate degree. 40% reported an annual household income of less than $60,000, 23% reported an income of $60,000-$100,000, and 37% reported a household income of greater than $100,000 per year. Slightly over half were married or living with a partner (54%). One in nine (11%) identified as gay, bisexual, or other self-described sexual orientation, while 89% were heterosexual

The majority of men (84%) reported normal or mild depression and/or anxiety symptoms in the past 2 weeks (23% reported mild symptoms, while 16% reported moderate or severe symptoms). Eight percent of men reported binge drinking six or more times in the prior month, and five percent reported illicit on non-prescribed prescription drug use six or more times in this time frame. One fifth (21%) reported that their neighborhood was 'extremely safe', with 52% considering their neighborhood 'quite safe' and 28% reporting that their neighborhood is 'slightly' or 'not at all' safe. Nearly one-third (29%) own a gun.

## Reported IPV perpetration

An estimated 282,865 Californian men (1.9% [95% CI 1.4–2.5%, N = 126]) reported IPV perpetration within the past year (see Table 1). This rate fell somewhat from 2021 (2.5%, 95% CI 1.5–4.3%) and 2022 (2.6%, 95% CI 1.7–4.0%) to 2023 (1.0%, 95% CI 0.6%-1.8%) (2022 vs 2023 p = 0.02). Physical IPV (including physical abuse and knife or gun violence) was reported by 0.9% of men (95% CI 0.6–1.4%, N = 77). Sexual IPV (including sexual harassment or assault) was reported by 1.6% of men (95% CI 1.1–2.2%, N = 89). Among men reporting perpetrating any form of violence in the past year (6.9% [95% CI 5.8–8.1%] of all men, n = 366), 27.3% (95%CI 20.6–35.2%) reported perpetrating violence against a partner.

## Gender equality beliefs

Respondents were mostly likely to endorse the item 'On the whole, men make better political leaders than women', with 20% of men indicating 'somewhat agree' or 'agree'. 11% endorsed the item 'A university education is more important for a boy than for a girl' and 16% endorsed the item 'On the whole, men make better business executives than women do'. The average scale score for the three items was 2.3 (SD 1.1) on a 1–5 scale, with higher score reflecting more unequal gender beliefs. The average score increased slightly, but significantly, over time, from 2.24 in 2021 to 2.38 in 2023.

**Table 1. Unweighted and weighted sample characteristics of men in CalVEX 2021–23 (n = 3609).**

|  | Unweighted N | Weighted % | Population estimate* |
|---|---|---|---|
| *N* | *3609* | *3609* | *15,126,455* |
| ***Any IPV*** | ***126*** | ***1.9%*** | ***282,865*** |
| *Any physical IPV* | *77* | *0.9%* | *136,138* |
| Physical abuse | 56 | 0.5% | 69,582 |
| Knife violence | 23 | 0.4% | 52,943 |
| Gun violence | 13 | 0.3% | 40,841 |
| *Any sexual IPV* | *89* | *1.6%* | *237,485* |
| Verbal Sexual Harassment (SH) | 48 | 0.9% | 140,676 |
| Homophobic/transphobic SH | 13 | 0.1% | 18,152 |
| Cyber SH | 23 | 0.3% | 39,329 |
| Physically aggressive SH | 20 | 0.4% | 58,993 |
| Quid pro quo/coercive SH | 6 | 0.1% | 7,563 |
| Forced sex (sexual assault) | 4 | 0.1% | 19,664 |

*Population estimates from Census ACS 2021 estimates, table S0101: https://data.census.gov/

## Bivariate associations with self-reported IPV perpetration

More unequal gender beliefs were significantly associated with self-reported IPV perpetration in bivariate comparisons; gender equality belief scores were 1.6 points higher (more unequal) among those who reported perpetrating IPV compared to those who did not (3.90 vs 2.30, p<0.001) (see Table 2). Endorsement of each individual attitude item was also independently associated with self-reported IPV perpetration in bivariate comparison.

Mental health symptoms, substance use, and gun ownership were all also significantly more frequently reported by men who reported perpetrating IPV than by men who did not.

## Unadjusted and adjusted associations with self-reported IPV perpetration

In unadjusted models, a one-point increase in the gender equality beliefs score was associated with four times greater odds of past-year self-reported IPV perpetration (AOR 3.97, 95% CI 2.96–5.33) (See Table 3). In models controlling for sociodemographic and additional factors of interest, more gender inequitable beliefs remained strongly and significantly associated with self-reported IPV perpetration. Accounting only for sociodemographic characteristics and year, a one-point increase in the gender equality beliefs scale score was associated with 3.15 times greater odds of self-reported past-year IPV perpetration (AOR 3.15, 95% CI 2.43–4.08). This association was attenuated somewhat when additionally accounting for mental health symptoms, substance use, neighborhood safety, and gun ownership (AOR 2.14, 95% CI 1.61–2.86). Slight further attenuation in effect was seen when additionally controlling for self-reported past year IPV victimization (AOR 1.85, 95% CI 1.38–2.49; results not shown). However, associations between gender inequitable beliefs and IPV perpetration remained strong and significant even when accounting for these additional related factors.

## Sensitivity analysis: Results stratified by sexual orientation

Past year IPV perpetration was reported by 1.6% of heterosexual men and 4.2% of GBO men (p = 0.01). Average gender equality beliefs score did not differ statistically significantly between heterosexual and GBO men (2.23 vs 2.35, p = 0.21). In unadjusted models, higher gender equality beliefs score was associated with increased odds of self-reported past year IPV perpetration

**Table 2. Bivariate associations with self-reported past-year IPV perpetration among California men 2021–23 (n = 3609).**

| Factor | Level | Overall | No past year IPV perpetration | Past year IPV perpetration | p-value |
|---|---|---|---|---|---|
| *N* | | *3609* | *3483* | *126* | |
| *Gender Equality Belief Scale* | | | | | |
| *Gender Equality Belief Scale* | *Continuous score; Range 1–5; Higher score = greater agreement; Mean (SD)* | *2.33 (1.12)* | *2.30 (1.09)* | *3.90 (1.23)* | *<0.001* |
| Item 1: On the whole, men make better political leaders than women | Disagree, somewhat disagree, neither agree nor disagree | 2781 (80.2%) | 2760 (81.3%) | 21 (23.0%) | <0.001 |
| | Agree or somewhat agree | 828 (19.8%) | 723 (18.7%) | 105 (77.0%) | |
| Item 2: A university education is more important for a boy than for a girl | Disagree, somewhat disagree, neither agree nor disagree | 3071 (89.3%) | 3042 (90.5%) | 29 (28.9%) | <0.001 |
| | Agree or somewhat agree | 538 (10.7%) | 441 (9.5%) | 97 (71.1%) | |
| Item 3: On the whole, men make better business executives than women do | Disagree, somewhat disagree, neither agree nor disagree | 2903 (83.9%) | 2874 (84.7%) | 29 (39.4%) | <0.001 |
| | Agree or somewhat agree | 706 (16.1%) | 609 (15.3%) | 97 (60.6%) | |
| *Sociodemographics* | | | | | |
| Age | 18–24 | 257 (8.9%) | 252 (8.9%) | 5 (5.0%) | <0.001 |
| | 25–34 | 561 (18.1%) | 515 (17.6%) | 46 (45.8%) | |
| | 35–44 | 802 (21.2%) | 754 (21.0%) | 48 (33.0%) | |
| | 45–54 | 503 (14.9%) | 478 (15.0%) | 25 (14.0%) | |
| | 55–64 | 610 (16.5%) | 609 (16.8%) | 1 (0.9%) | |
| | 65–74 | 559 (13.2%) | 558 (13.5%) | 1 (1.3%) | |
| | 75+ | 317 (7.2%) | 317 (7.3%) | 0 (0.0%) | |
| Race/Ethnicity | White | 1790 (41.0%) | 1716 (41.1%) | 74 (37.9%) | 0.001 |
| | Hispanic | 733 (32.6%) | 695 (32.2%) | 38 (52.9%) | |
| | Black, Asian, Other/multiple races | 1086 (26.4%) | 1072 (26.7%) | 14 (9.2%) | |
| Ideology | Liberal | 1063 (28.3%) | 1001 (27.7%) | 62 (60.1%) | <0.001 |
| | Moderate | 1672 (48.0%) | 1629 (48.4%) | 45 (31.5%) | |
| | Conservative | 843 (23.7%) | 823 (23.9%) | 20 (8.4%) | |
| Education | Less than Highschool (HS) | 115 (11.1%) | 144 (10.9%) | 11 (26.4%) | 0.002 |
| | Completed HS/some college/Advanced degree | 1464 (47.4%) | 1446 (47.8%) | 18 (26.5%) | |
| | Bachelor Degree/4yr college degree | 1000 (22.2%) | 967 (22.3%) | 33 (15.7%) | |
| | Graduate degree | 990 (19.3%) | 926 (19.0%) | 64 (31.4%) | |

*(Continued)*

**Table 2.** (Continued)

| Factor | Level | Overall | No past year IPV perpetration | Past year IPV perpetration | p-value |
|---|---|---|---|---|---|
| Household Income | Less than $60,000 | 1295 (39.7%) | 1280 (39.9%) | 15 (30.0%) | 0.47 |
| | $60,000 to under $100,000 | 829 (23.4%) | 806 (23.3%) | 23 (27.1%) | |
| | $100,000 or more | 1485 (36.9%) | 1397 (36.8%) | 88 (42.9%) | |
| Marital Status | Not married or living with partner | 1498 (46.3%) | 1483 (46.5%) | 15 (33.3%) | 0.14 |
| | Married or living with partner | 2108 (53.7%) | 1997 (53.5%) | 111 (66.7%) | |
| Sexual Orientation | Gay, Bisexual/other | 414 (10.8%) | 395 (10.6%) | 19 (24.2%) | 0.01 |
| | Heterosexual | 3190 (89.2%) | 3083 (89.4%) | 107 (75.8%) | |
| Year | 2021 | 1153 (27.3%) | 1099 (27.1%) | 54 (36.8%) | 0.03 |
| | 2022 | 1004 (28.4%) | 955 (28.2%) | 49 (39.0%) | |
| | 2023 | 1452 (44.3%) | 1429 (44.7%) | 23 (24.2%) | |
| *Additional variables of interest* | | | | | |
| Depression and/or anxiety symptoms | Normal/mild | 2878 (84.1%) | 2837 (84.8%) | 41 (46.4%) | <0.001 |
| | Moderate/severe | 678 (15.9%) | 597 (15.2%) | 81 (53.6%) | |
| 6+ days of binge drinking in past month | No | 3243 (92.5%) | 3174 (93.1%) | 69 (59.4%) | <0.001 |
| | Yes | 361 (7.5%) | 304 (6.9%) | 57 (40.6%) | |
| 6+ days of illicit drug/non-prescription drug use in past month | No | 3328 (94.6%) | 3255 (95.1%) | 73 (63.8%) | <0.001 |
| | Yes | 271 (5.4%) | 220 (4.9%) | 51 (36.2%) | |
| How safe do you think your neighborhood is from violence and crime? | Extremely safe | 911 (20.6%) | 840 (20.3%) | 71 (34.4%) | 0.06 |
| | Quite safe | 1834 (51.7%) | 1798 (52.0%) | 36 (36.5%) | |
| | Slightly/Not at all safe | 857 (27.7%) | 838 (27.6%) | 19 (29.1%) | |
| Current gun ownership | No | 2384 (70.9%) | 2370 (72.0%) | 14 (15.4%) | <0.001 |
| | Yes | 1192 (29.1%) | 1083 (28.0%) | 109 (84.6%) | |

for both heterosexual men (OR 3.92, 95% CI 2.89–5.33) and GBO men (OR 3.60, 95% CI 1.83–7.09) [See S1 Table]. These associations remained after controlling for sociodemographic characteristics (heterosexual AOR 3.10, 95% 2.29–4.20; GBO AOR 3.38, 95% CI 2.01–5.69).

## Discussion

Results offer a sound, though likely conservative, population estimate of men's self-reported IPV perpetration in the past year and highlight a strong association between holding more

**Table 3. Unadjusted and adjusted models exploring the association between gender equality beliefs and self-reported IPV perpetration among Californian men 2021–2023 (n = 3609).**

| | M1: Unadjusted | | M2: Socio-demographics | | M3: M2 + mental health, neighborhood violence, gun ownership | |
|---|---|---|---|---|---|---|
| | OR | 95%CI | AOR | 95%CI | AOR | 95%CI |
| Gender equality beliefs scale score | 3.97 | 2.96,5.33 | 3.15 | 2.43–4.08 | 2.14 | 1.61–2.86 |
| Age, Years, continuous | – | – | 0.97 | 0.95–0.99 | 0.98 | 0.95–1.00 |
| Race/ethnicity | | | | | | |
| White | – | – | Ref | Ref | Ref | Ref |
| Hispanic | – | – | 1.07 | 0.54–2.12 | 1.07 | 0.51–2.26 |
| Black, Asian, Other/multiple race | – | – | 0.40 | 0.17–0.92 | 0.52 | 0.19–1.42 |
| Political ideology | | | | | | |
| Liberal | – | – | Ref | Ref | Ref | Ref |
| Moderate | – | – | 0.40 | 0.17–0.95 | 0.53 | 0.21–1.37 |
| Conservative | – | – | 0.13 | 0.06–0.30 | 0.15 | 0.06–0.38 |
| Income | | | | | | |
| Less than 60k | – | – | Ref | Ref | Ref | Ref |
| 60k-<100k | – | – | 1.80 | 0.72–4.52 | 1.78 | 0.60–5.32 |
| 100k+ | – | – | 1.75 | 0.78–3.91 | 1.23 | 0.45–3.35 |
| Currently married | | | | | | |
| No | – | – | Ref | Ref | Ref | Ref |
| Yes | – | – | 1.43 | 0.65–3.15 | 1.37 | 0.57–3.28 |
| Sexual Orientation | | | | | | |
| Gay, Bisexual, Other | – | – | Ref | Ref | Ref | Ref |
| Heterosexual | – | – | 0.55 | 0.19–1.55 | 0.38 | 0.14–1.07 |
| Year | | | | | | |
| 2021 | – | – | Ref | Ref | Ref | Ref |
| 2022 | – | – | 1.01 | 0.43–2.37 | 1.05 | 0.40–2.77 |
| 2023 | – | – | 0.46 | 0.18–1.21 | 0.44 | 0.15–1.33 |
| Mental health symptoms | | | | | | |
| Normal/Mild | – | – | – | – | Ref | Ref |
| Moderate/Severe | | | | | 2.27 | 0.91–5.69 |
| 6+ days of binge drinking in past month | | | | | | |
| No | – | – | – | – | Ref | Ref |
| Yes | – | – | – | – | 3.55 | 1.46–8.62 |
| 6+ days of drug use in past month | | | | | | |
| No | – | – | – | – | Ref | Ref |
| Yes | – | – | – | – | 2.07 | 0.89–4.83 |
| Neighborhood safety | | | | | | |
| Extremely safe | – | – | – | – | Ref | Ref |
| Quite safe | – | – | – | – | 1.17 | 0.48–2.84 |
| Slightly/not at all safe | – | – | – | – | 1.46 | 0.50–4.26 |
| Owns a gun | – | – | – | – | | |
| No | – | – | – | – | Ref | Ref |
| Yes | – | – | – | – | 6.48 | 2.55–16.48 |

gender inequitable beliefs and self-reported IPV perpetration among men. We find that approximately 1 in every 50 adult men in California—or 282,865 California men–perpetrated IPV in the past year. These findings offer a first time estimate of self-reported IPV perpetration at a state level within the U.S. Survey data reports on men's IPV perpetration with a population representative sample are not common. Prior evidence indicate that this prevalence estimate is within the realm of population-based estimates of men's IPV perpetration in Eastern Europe, but much lower than that seen in the Asia-Pacific region [48], suggesting geographic variation in this variable is likely. We also see some variation in the form of IPV perpetration reported [49]. We found higher reporting of sexual IPV perpetration compared with physical IPV perpetration (1.6% vs 0.9%) in our sample, likely because our assessment of sexual IPV included harassment. Verbal sexual harassment was the most commonly reported form of IPV perpetration. These findings overall indicate that IPV perpetration assessments do yield reports, and even if estimates are conservative, offer insight into the nature of this perpetration.

Our findings, which pooled data from across three years which included COVID, indicate that reports of IPV perpetration were slightly lower in 2023 than 2022 and 2021. These data suggest a decline in perpetration as the pandemic severity dissipated, a finding that corresponds with reports on violence against women in California in this same timeframe [34]. Nonetheless, regardless of year, men endorsing gender inequitable beliefs were more likely to report IPV perpetration in the past year. More than three in five men who have perpetrated violence against their intimate partners in the last 12 months agreed or strongly agreed to gender inequitable beliefs about women in leadership. These findings are in line with other research conducted among young men in the U.S., that men reporting more traditional gender beliefs have nearly two-times greater odds of reporting IPV perpetration in the past year [46], a findings also seen in other national contexts [17, 20, 45, 50]. However, this is to our knowledge the first study that documents at scale the role men's beliefs regarding women's equality may have on their self-reported perpetration of IPV.

Measures of gender inequitable attitudes, norms, and beliefs vary widely and differ in their associations with IPV perpetration among men [26]. Counter to prior research conducted across the globe, our findings suggest that even distal views of unequal gender beliefs surrounding women's roles in leadership outside the home are strongly associated with IPV perpetration, even after adjusting for important drivers of violence such as mental health, alcohol and drug use, gun ownership, and experiences of violence. Previous research using ecological level data has shown that nations with more gender unequal beliefs regarding women's advancement in society tend to have less women in leadership positions [51, 52]. While our study showed that gender unequal views are not common at the population level (less than 20% of men agreed or somewhat agreed to gender role perception items), our data expands on these global studies to demonstrate that Californian men who hold these views are also significantly more likely to perpetrate violence against their partners, regardless of sexual orientation.

## Impact of substance use and mental health on findings

Bivariate results found that over a third of men who reported perpetrating IPV in the past year binge drank and/or used illicit substances more than six times in the month prior to completing the survey compared to less than 10% of men who did not report perpetrating IPV. Also, over half of men reporting IPV perpetration in the past year had moderate to severe depression and/or anxiety symptoms compared to only 15% of men who did not report perpetrating IPV. These findings are in line with a well-established body of literature that links substance use and mental health with elevated rates and consequences of IPV [9–12, 44, 53]. Our findings

highlight that even when adjusting for high levels of depression, anxiety, alcohol and illicit drug use, the association between unequal gender role beliefs and self-reported IPV perpetration among Californian men remained. Thus, while programs and policies that aim to treat substance use disorders among men or reduce alcohol and drug use may be beneficial at reducing IPV [54], efforts need to consider how individual and societal level gender equality beliefs towards women's advancement can be addressed simultaneously with issues around substance use as well as depression and anxiety.

## Impact of gun ownership on association

The association between gender equality beliefs and self-reported IPV perpetration was reduced after adjusting for gun ownership, an important covariate of IPV perpetration [55] that has also been associated with gender unequal beliefs towards women [56]. Bivariate results showed that 85% of men who reported perpetrating IPV owned a gun compared to only 28% of men not reporting IPV perpetration. These findings have serious implications for femicide rates in California, as US data indicates that abusers who have access to a gun are five times more likely to kill their partners [57–59]. These staggering findings highlight that even in California, where gun ownership is lower and gun control laws are stricter than many other states, almost all men who reported IPV perpetration in the past year owned a gun. While only 0.3% of men reported perpetrating gun violence against their partner, evidence suggests that men that commit gun-related homicide are much more likely to have a history of IPV perpetration [60]. These findings highlight the critical need for increased gun control among men with histories of domestic abuse [59]. In the U.S., gun ownership restrictions among violators of domestic violence offenders have shown efficacy in reducing rates of IPV-related homicides, while similar policies aimed at alcohol taxation as a means of reducing IPV-related homicides have not been as effective [61, 62]. While there is some progress being made towards policies aimed at reducing and preventing gun ownership among men with a history of domestic violence, additional efforts are critically needed to prevent abusers from owning guns and requiring abusers to relinquish guns they already have.

## Implications of findings

Our findings document that men's beliefs regarding the value and capacity of women in society relate to their self-reported perpetration of IPV, extending prior work on masculinity beliefs and IPV to include men's beliefs regarding the societal value of women and IPV. Importantly, we see these effects even after adjusting for known key drivers of IPV–mental health, substance use, and gun ownership, factors not only adjusted for in our model but also showing, as seen in prior work, a strong association with men's self-reported IPV perpetration. These findings highlight the integrated behavioral and ideologic risks that often co-exist in ways that can increase men's violence against women. Consequently, they also suggest the importance of multi-level IPV prevention efforts. These should include programmatic support for mental and emotional well-being, policy supports that can impede gun access for those who may be a risk to themselves or others, and social norm change with regard to men's value and respect for women in leadership. Global and national escalation of violence against women in public life reinforce the need for focus on this concern [63].

A growing body of literature has explored the use of gender transformative approaches to support men in shifting inequitable gender role perceptions to reduce IPV perpetration [64]. These efforts are grounded in theoretical understandings that efforts to reduce violence at a population level need to engage men and boys to cultivate equitable attitudes, beliefs, and in turn relationships with families and social networks, raise awareness about violence, and

become change agents for more equitable communities [65]. Much of the research showing success of gender transformative approaches have been conducted in small, non-representative samples with minimal follow-up time, highlighting the need to scale up of approaches that work to shift gender role perceptions at a population level [66]. They also tend to focus on gender roles and expectation in domestic life. Our results further expand upon the need to work through perceptions of IPV justification and household gender-based power dynamics to include programming that addresses men's broader biases towards women in power and leadership roles outside the home.

## Limitations

The results herein should be taken in light of several limitations. First, our sample only includes cisgender men; transgender identity was not consistently assessed over the survey years, and too few individuals indicated transgender identity in 2021 and 2022 to examine cisgender and transgender populations separately. Additionally, while we present findings stratified by sexual orientation, we did not have sufficient power to explore associations between gender equality beliefs and perpetration separately for heterosexual and non-heterosexual men in greater detail. Given the strong association between gender unequal beliefs and IPV perpetration among GBO in CalVEX, additional efforts are needed to explore how gender equality beliefs impact IPV perpetration and experiences among gender and sexual minority men. Next, while the weighted survey samples are representative of the adult state population with regards to a number of sociodemographic factors, the sample does exclude institutionalized populations (such as those in prisons, jails, nursing homes, or other long-term medical facilities) and state residents who do not speak English or Spanish; our results are thus not generalizable to these populations. While we adjusted for several key confounding factors in our analysis, there are unmeasured factors that we were unable to adjust for, thus our analysis may be subject to some residual confounding bias. For example, we only had data on witnessing of IPV in childhood for the 2023 survey. We know that this is an important determinant of both gender role perceptions and IPV perpetration and we did find a strong bivariate association between witnessing violence as a child and self-reported IPV perpetration in 2023. In 2023 data, men who reported perpetrating IPV were more likely to have witnessed it as a child: 57% of men who reporting perpetrating IPV in the past year witnessed it as a child, while 13% of men who did not report perpetrating IPV witnessed IPV as a child (p<0.001). The reported level of IPV perpetration was quite low (less than 2%), and lower than what would be expected based on reports from women regarding victimization from IPV [34, 35]. IPV perpetration is often underreported in survey research due to social desirability and may have been especially under-reported in this study [67, 68], as we used a two-step approach to asking about IPV perpetration, whereby the perpetrator first had to report perpetrating any of nine named forms of violence, and then subsequently indicate who they perpetrated the violence against. This measure thus only captures physical and sexual violence perpetration and does not capture other forms of IPV such as emotional, financial, or controlling forms of violence against a partner. Finally, while the same men may possibly be surveyed in multiple years, we did not have unique identifiers across survey waves and thus could not account for repeated observations from the same individual. All observations are thus considered unique individuals in regression analyses.

## Conclusion

Results from this study highlight that Californian men with gender inequitable beliefs surrounding women's role in leadership are significantly more likely to have reported

perpetrating violence in the last 12 months. While gender inequitable views towards women in leadership and self-reported IPV perpetration were reported among a minority of men in California, these same men were more likely to report owning a gun, have moderate to severe anxiety and/or depression, drink heavily, and use substances many times per month, signifying a need for gender transformative efforts and policies that support treatment, work with men to shift perceptions to be more equitable towards women, and reduce the likelihood of men with a history of domestic violence possessing guns.

## Supporting information

**S1 Table. Sexual orientation stratified.**
(XLSX)

**S1 File. IPV perpetration and gender beliefs manuscript.**
(PDF)

## Acknowledgments

We would like to that the following individuals/community partners for their support: The Blue Shield of California Foundation, Kaiser Permanente, The East Bay Community Foundation, VALOR US, AAPI Data, NORC at the University of Chicago, and The Reis Group, the California Department of Public Health. We would also like to thank the participants of this study for graciously sharing their personal experiences.

## Author Contributions

**Conceptualization:** Kalysha Closson.

**Formal analysis:** Nicole E. Johns.

**Project administration:** Nicole E. Johns.

**Supervision:** Anita Raj.

**Writing – original draft:** Kalysha Closson, Nicole E. Johns, Anita Raj.

**Writing – review & editing:** Kalysha Closson, Nicole E. Johns, Anita Raj.

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
