## [Decision Letter · Decision Letter 0]

12 Oct 2024

PONE-D-24-13633Are men’s gender equality beliefs surrounding women in leadership associated with intimate partner violence perpetration? A state-level analysis of California menPLOS ONE

Dear Dr. Closson,

Thank you for submitting your manuscript to PLOS ONE. After careful consideration, we feel that it has merit but does need some additional work. In fact, both reviewers recommend minor revisions. Therefore, we invite you to submit a revised version of the manuscript that addresses the points raised during the review process.

The reviewers both agree that the manuscript has merit, but it lacks a little bit of clarity. First and foremost, Reviewer #1 is stressing the need to tone done any instances of causal language. I most definitely agree with the reviewer. I also agree with this reviewer that a key part of the revision of the manuscript involves improving the discussion of the representativeness of the sample and the generalizability of the results. Both reviewers ask to extend the analysis in two ways. Reviewer #2 is asking for additional robustness analysis. Hopefully, this is feasible. Reviewer #1 would like to see the analysis to be extended to other forms of violence: emotional and psychological, among others. I think this would be interesting, especially for these two.

When revising your manuscript, please consider all issues mentioned in the reviewers' comments carefully: please outline every change made in response to their comments and provide suitable rebuttals for any comments not addressed. The reviewers have provided you with some constructive and careful comments. I look forward to seeing your response. Please note that your revised submission may need to be re-reviewed.

Thank you again for the opportunity to read this very interesting paper, and I hope the comments and suggestions provided by the referees will prove helpful to you in revising the paper.

We look forward to receiving your revised manuscript.

Kind regards,

Michele Baggio

Academic Editor

PLOS ONE

“This work was supported by Blue Shield of California Foundation (https://blueshieldcafoundation.org) #COV-2207-18216 (AR) and Bill and Melinda Gates Foundation (https://www.gatesfoundation.org/) #INV002967 (AR). The funders had no role in the study design, data collection or analysis, decision to publish, or preparation of the manuscript.”

3. Please note that your Data Availability Statement is currently missing [the repository name and/or the DOI/accession number of each dataset OR a direct link to access each database]. If your manuscript is accepted for publication, you will be asked to provide these details on a very short timeline. We therefore suggest that you provide this information now, though we will not hold up the peer review process if you are unable.

Reviewers' comments:

Reviewer's Responses to Questions

**Comments to the Author**

1. Is the manuscript technically sound, and do the data support the conclusions?

Reviewer #1: Yes

Reviewer #2: Partly

2. Has the statistical analysis been performed appropriately and rigorously? 

Reviewer #1: Yes

Reviewer #2: Yes

3. Have the authors made all data underlying the findings in their manuscript fully available?

Reviewer #1: Yes

Reviewer #2: Yes

4. Is the manuscript presented in an intelligible fashion and written in standard English?

Reviewer #1: Yes

Reviewer #2: Yes

5. Review Comments to the Author

Reviewer #1: See attached referee report. The paper meets the criteria outlined above. The authors should be careful to avoid causal language and limit their analysis to self-reported IPV perpetration rather than generalizing this to all IPV perpetration.

Reviewer #2: Review: PLOS ONE

“Are men’s gender equality beliefs surrounding women in leadership associated with intimate partner violence perpetration? A state-level analysis of California men”

The paper investigates how men’s beliefs around women in leadership are associated with IPV, and find a strong and persistent association between gender-traditional beliefs and IPV, even after controlling for factors such as mental health and gun ownership. They use interesting new data on violence experience and conduct careful statistical analysis. I think the paper makes a nice contribution to the literature and have some comments and suggestions for a revision:

Main comments:

1. My biggest comment is why use chose to use only those three gender-related items from the WVS? How about “When jobs are scarce a man should have more right to a job than a woman”? Or perhaps the questions on if housewifery is just a fulfilling or if a child suffers when the mother is working? I lack additional explanations for why these three items were selected and not the other ones (the text says “expert opinion” and discussions with CalVeX), and I would have liked to see some robustness analysis using the other items or a broader measure containing all the gender equality-related items.

2. Related, in the text you write about gender norms or gender equality beliefs, but the title talks about beliefs on female leadership. I think the authors should be coherent in how they talk about the norms they are capturing. And I don’t see why norms about female leadership might be more important for IPV than gender norms in general?

3. What share of the total committed violence is against a spouse or romantic partner? I would like to know this number.

4. Can we look at physical and sexual violence separate and see if the association differs? I mean as an additional analysis: splitting the outcome in physical and sexual violence and see if any of the two are the one driving the association?

5. Age groups, why do you use only 18-29 and 30+? Might mask important heterogeneity as 82% are in 30+ group. I would like to see the results when age is separated in more age groups. Related to that, I would like to see the coefficients for the socio-demographic characteristics like age, race, education, marital status in the results table (table 3). It will become large, but as this is the most important table I think it should be allowed the space it needs.

Very minor comments:

1. On line 20 the word “better” seems to be missing. “Men make better political leaders”

2. On line 344 the word “be” seems to be missing. “Those who may be a risk”

6. PLOS authors have the option to publish the peer review history of their article (what does this mean?). If published, this will include your full peer review and any attached files.

Reviewer #1: **Yes: **Kaitlyn Sims

Reviewer #2: No

---

## [Author Response · Author response to Decision Letter 0]

4 Nov 2024

Response to reviewers PONE-D-24-13633

CALVEX gender beliefs 

RESPONSE: We have reviewed the above documents and have formatted our manuscript according to PLOS style requirements 

“This work was supported by Blue Shield of California Foundation (https://blueshieldcafoundation.org) #COV-2207-18216 (AR) and Bill and Melinda Gates Foundation (https://www.gatesfoundation.org/) #INV002967 (AR). The funders had no role in the study design, data collection or analysis, decision to publish, or preparation of the manuscript.”

RESPONSE: Please change the statement to the following (we have also updated this in the cover letter): 

“This work was supported by Blue Shield of California Foundation (https://blueshieldcafoundation.org) #COV-2207-18216 (AR) and Bill and Melinda Gates Foundation (https://www.gatesfoundation.org/) #INV002967 (AR). The funders had no role in the study design, data collection or analysis, decision to publish, or preparation of the manuscript. There was no additional external funding received for this study”

3. Please note that your Data Availability Statement is currently missing [the repository name and/or the DOI/accession number of each dataset OR a direct link to access each database]. If your manuscript is accepted for publication, you will be asked to provide these details on a very short timeline. We therefore suggest that you provide this information now, though we will not hold up the peer review process if you are unable.

Please see below the Data Availability Statement for this analysis: 

The data used for this study is publicly available via the Open ICPSR repository. 

o 2021: https://www.openicpsr.org/openicpsr/project/204402/version/V1/view

o 2022: https://www.openicpsr.org/openicpsr/project/204401/version/V1/view

o 2023: https://www.openicpsr.org/openicpsr/project/199087/version/V1/view

The code for this analysis is available in the supplementary files

RESPONSE: It seems that I have two PLoS ONE accounts and have my other account linked to my ORCID ID which is: https://orcid.org/0000-0002-2985-0610 Is there a way to help with merging the accounts? 

RESPONSE: We have reviewed the reference list and there are currently no articles needing to be retracted 

Reviewer 1: 

Referee report

This study uses survey data from California to test for an association between men’s own beliefs on gender equality and their self-reported IPV perpetration in the past year. In doing so, this study expands on prior work that describes society-level patriarchal values and IPV outcomes to instead consider how individual beliefs correspond to individual perpetration. The authors find that, among the 2% of their sample that self-reports perpetrating physical and sexual violence against an intimate partner in the past year, views supporting gender inequality are more prevalent. They further provide evidence that these correlate to policy-relevant characteristics such as gun ownership and substance abuse. 

The paper is clearly written and conveys a cohesive and coherent argument. My comments below are intended to strengthen the manuscript as written rather than change anything the authors are doing. Major/substantive comments should improve the work’s clarity, avoid overstating conclusions, and address common reader questions and concerns. The paper should also be careful to remove any instances of causal language for this purely associative/correlative study.

• RESPONSE: Thank you for taking the time to review our study. We appreciate the reviewer’s feedback and have done our best to address the reviewer comments, which have improved our manuscript. 

Major/substantive comments

• Be sure to carefully distinguish IPV perpetration from IPV perpetration self-reporting. My first instinct upon seeing that 2% of the sample—while this does reflect a large number of individuals overall—is that this feels like an underestimate. Much literature discusses the willingness to self-report risky and harmful behaviors. How does the selection in who in your sample is willing to self-report perpetrating IPV impact your findings? I recommend both engaging with this literature, discussing what it means for your sample and generalizability, and tempering language when discussing perpetration (in favor of self-reported perpetration).

o RESPONSE: We agree with the reviewer that this is most likely an underestimate of IPV perpetration. We have gone through the manuscript and added self-reported IPV perpetration, where appropriate. We also highlight this limitation in reporting bias in the limitation section as highlighted below. 

“IPV perpetration is often underreported in survey research due to social desirability and may have been especially under-reported in this study, as we used a two-step approach to asking about IPV perpetration, whereby the perpetrator first had to report perpetrating any of nine named forms of violence, and then subsequently indicate who they perpetrated the violence against.”

• We have added the following citations to the statement above: 

o Caetano, R., C. Field, S. Ramisetty-Mikler and S. Lipsky (2008). "Agreement on Reporting of Physical, Psychological, and Sexual Violence Among White, Black, and Hispanic Couples in the United States." Journal of Interpersonal Violence 24(8): 1318-1337.

o Follingstad, D. R. and M. J. Rogers (2013). "Validity concerns in the measurement of women’s and men’s report of intimate partner violence." Sex roles 69: 149-167.

• Why do you limit IPV perpetration to physical and sexual violence? Does CalVEX include information about emotional, psychological, economic, spiritual, etc. abuse? These more poorly understood forms of violence may be less subject to self-reporting bias compared to IPV that corresponds more directly to illegal behavior (assault, sexual assault, etc.).

o RESPONSE: In CalVEX we only asked about physical and sexual IPV perpetration and thus while we agree with the reviewer that other forms of IPV are important to consider, we do not have the data to explore this association. This is something that we plan to investigate further in future years of the CalVEX survey. 

• I would appreciate more discussion of the choice of additional variables of interest. It’s unclear to me (though I could be easily convinced!) why perceptions of neighborhood safety might be associated with either gender equality beliefs or IPV perpetration.

o RESPONSE: We agree with the reviewer and have added an additional sentence in the introduction talking about neighborhood and IPV perpetration: 

“Ecological factors, including the neighborhood environment and perceptions of one’s neighborhood have also been highlighted as important determinants of IPV [15]”.

• Line 364: you say that you analyze IPV perpetration association separately for non-heterosexual men in your sample and find differences but do not report results. While only 11% of the sample is non-heterosexual, this is still a substantial number that could be interpreted at least via descriptives, if not rigorously with statistical significance due to statistical power concerns. There is a strong argument to be made that, while queer/LGBTQ IPV is still connected to patriarchy (see Messinger, 2017), it is still fundamentally different when violence generated by patriarchal values is perpetrated by men against women versus by men against other men.

o We agree with the reviewer that this is an important topic that needs to be further explored and have discussed it in the limitations. We have added the following analysis to the methods and report on the results as noted below. 

“As a post-hoc sensitivity analysis, we examine self-reported IPV perpetration stratified by sexual identity, e.g. separately examine rates of perpetration and gender equality beliefs scale score among heterosexual vs gay, bisexual, or another non-heterosexual sexual identity (GBO) men. We also replicate unadjusted and sociodemographic-adjusted models examining the association between self-reported perpetration and gender equality beliefs, stratified by sexual identity.”

Sensitivity analysis: results stratified by sexual identity 

Past year IPV perpetration was reported by 1.6% of heterosexual men and 4.2% of GBO men (p=0.01). Average gender equality beliefs score did not differ statistically significantly between heterosexual and GBO men (2.23 vs 2.35, p=0.21). In unadjusted models, higher gender equality beliefs score was associated with increased odds of self-reported past year IPV perpetration for both heterosexual men (OR 3.92, 95% CI 2.89-5.33) and GBO men (OR 3.60, 95% CI 1.83-7.09) [See Appendix Table 1]. These associations remained after controlling for sociodemographic characteristics (heterosexual AOR 3.10, 95% 2.29-4.20; GBO AOR 3.38, 95% CI 2.01-5.69). 

• The authors should demonstrate a lack of multicollinearity in the included demographic factors and additional variables of interest, perhaps via a correlation matrix.

o RESPONSE: We ran a test for multicollinearity on the demographic factors included in the model and found some evidence of multicollinearity for education, and thus have removed it from the model. The following details have been added to the methods: 

“All sociodemographic factors noted above as well as mental health symptoms, substance use, neighborhood safety, and gun ownership were considered for inclusion in adjusted models; due to evidence of multicollinearity (VIF >6), education was removed; VIFs were <2 for all remaining factors and thus all other variables were retained in adjusted models.”

• Please present p-values/confidence intervals backing up the claim that reductions in perpetration are decreasing year over year.

o RESPONSE: P-values and confidence intervals for reductions in perpetration can be found in the “Reported IPV perpetration” section of the results in the following sentence: 

An estimated 282,865 Californian men (1.9% [95% CI 1.4-2.5%, N=126]) reported IPV perpetration within the past year (see Table 1). This rate fell somewhat from 2021 (2.5%, 95% CI 1.5-4.3%) and 2022 (2.6%, 95% CI 1.7-4.0%) to 2023 (1.0%, 95% CI 0.6%-1.8%) (2022 vs 2023 p=0.02).

• Line 68-69: “Yet, we lack data in understanding if men’s views of women and gender equality are actually associated with IPV perpetration.” I think this claim should be tempered—we do know a lot about the role of patriarchy in generating patterns of DV and IPV, and this connection has been central to the movement fighting violence against women for decades. However, this is an aggregate and society-level connection. Perhaps it is more appropriate to say that we know less about how individual men’s views on gender equality correspond to individual perpetration of IPV in men’s own intimate relationships.

o RESPONSE: We thank the reviewer for this suggestion and have changed this claim so that it now states: 

“Thus, while we have data understanding this connection at a society-level connection, we lack data in understanding how individual men’s views of women and gender equality correspond with individual perpetration of IPV”

• On line 197 you state “no men declined to answer the IPV perpetration question,” but the subsequent table 1 includes a number of IPV metrics. How was this question structured if it is a single question but elicited this much detail on individual forms of IPV perpetration?

o RESPONSE: We apologize for the confusion. We asked a series of questions related to physical and sexual IPV, if respondents answered yes to any of the forms of violence perpetration, they were then asked who violence was perpetrated against for each form of violence they perpetrated. We apologize if our exclusion criteria for IPV was unclear. We have updated the results so that it now states: 

“no men declined to answer the set of items from which we derived self-reported IPV perpetration”

• Please remove all causal language from the manuscript (e.g., lines 315 and 336). Your methods are descriptive/correlative and should not be interpreted causally.

o RESPONSE: We thank the reviewer for raising this important point. We have gone through the manuscript and removed any casual language. 

Minor comments

• Review the piece for grammar and spelling throughout

o RESPONSE: We have reviewed the manuscript for grammar and spelling throughout 

• Be sure to spell out any acronyms at their first instance.

o We have reviewed the manuscript and ensured that we have spelled out all acronyms at first use. Note that NORC is not an acronym 

• Some sentences are far too long (e.g., line 56-61) to be easily parsed. 

o We have reviewed the manuscript and broke up lengthy sentences

• Line 295-297 is unclear.

o RESPONSE: We have revised the sentence in hopes that it is now clearer: 

“Previous research using ecological level data has shown that nations with more gender unequal beliefs regarding women’s advancement in society tend to have less women in leadership positions [51, 52]”

• Does CalVEX include data on violence perpetrated against former intimate/sexual partners? 

o RESPONSE: For each form of violence asked about (in the past 12 months), respondents were asked to then answer who they perpetrated the violence against with the following options: 

An adult family member, r

---

## [Editor Report · Decision Letter 1]

25 Nov 2024

Are men’s gender equality beliefs associated with self-reported intimate partner violence perpetration? A state-level analysis of California men

PONE-D-24-13633R1

Dear Dr. Closson,

We’re pleased to inform you that your manuscript has been judged scientifically suitable for publication and will be formally accepted for publication once it meets all outstanding technical requirements.

Kind regards,

Michele Baggio

Academic Editor

PLOS ONE

---

## [Editor Report · Acceptance letter]

26 Nov 2024

PONE-D-24-13633R1 

PLOS ONE

Dear Dr. Closson, 

I'm pleased to inform you that your manuscript has been deemed suitable for publication in PLOS ONE. Congratulations! Your manuscript is now being handed over to our production team.

Kind regards, 

on behalf of

Dr. Michele Baggio 

Academic Editor

PLOS ONE